# Determination of Complex Formation between Drosophila Nrf2 and GATA4 Factors at Selective Chromatin Loci Demonstrates Transcription Coactivation

**DOI:** 10.3390/cells12060938

**Published:** 2023-03-19

**Authors:** Emma Neidviecky, Huai Deng

**Affiliations:** Department of Biology, University of Minnesota Duluth, 1035 Kirby Drive, Duluth, MN 55812, USA

**Keywords:** Nrf2, CncC, GATA4, Pannier, BiFC, polytene chromosome, development, *Drosophila*

## Abstract

Nrf2 is the dominant cellular stress response factor that protects cells through transcriptional responses to xenobiotic and oxidative stimuli. Nrf2 malfunction is highly correlated with many human diseases, but the underlying molecular mechanisms remain to be fully uncovered. GATA4 is a conserved GATA family transcription factor that is essential for cardiac and dorsal epidermal development. Here, we describe a novel interaction between *Drosophila* Nrf2 and GATA4 proteins, i.e., cap‘n’collar C (CncC) and Pannier (Pnr), respectively. Using the bimolecular fluorescence complementation (BiFC) assay—a unique imaging tool for probing protein complexes in living cells—we detected CncC–Pnr complexes in the nuclei of *Drosophila* embryonic and salivary gland cells. Visualization of CncC–Pnr BiFC signals on the polytene chromosome revealed that CncC and Pnr tend to form complexes in euchromatic regions, with a preference for loci that are not highly occupied by CncC or Pnr alone. Most genes within these loci are activated by the CncC–Pnr BiFC, but not by individually expressed CncC or Pnr fusion proteins, indicating a novel mechanism whereby CncC and Pnr interact at specific genomic loci and coactivate genes at these loci. Finally, CncC-induced early lethality can be rescued by Pnr depletion, suggesting that CncC and Pnr function in the same genetic pathway during the early development of *Drosophila*. Taken together, these results elucidate a novel crosstalk between the Nrf2 xenobiotic/oxidative response factor and GATA factors in the transcriptional regulation of development. This study also demonstrates that the polytene chromosome BiFC assay is a valuable tool for mapping genes that are targeted by specific transcription factor complexes.

## 1. Introduction

Nrf2 (NF-E2-related factor 2) is a transcription factor of the cap’n’collar (cnc) subfamily that plays an essential role in cell protection by mediating transcriptional responses to environmental toxins (xenobiotics) and oxidative stimuli [1,2]. Nrf2 can bind to antioxidant response elements (AREs) and activate a series of antioxidant and detoxifying genes [3,4,5]. Under basal conditions, Nrf2 is inhibited by Keap1 (Kelch-like ECH-associated protein 1)—an E3 adaptor protein that binds to Nrf2 in the cytoplasm and mediates its proteasomal degradation. Xenobiotic and oxidative compounds disrupt the Keap1–Nrf2 interaction, releasing Nrf2 to enter the nucleus and activate target genes [1,6].

Mutations that misregulate Nrf2 are associated with many diseases, including cancer [7], respiratory diseases [8], neurodegeneration [9], and cardiovascular diseases [10]. However, the complete range of roles that Nrf2 plays in both pathology and physiology remains to be fully identified. Recent studies in several model systems have demonstrated that Nrf2 can directly target and regulate developmental genes [11,12]. For example, murine Nrf2 can control adipogenesis through activating adipogenetic genes [13,14,15], promote cell proliferation via the transcriptional activation of glucose metabolic enzymes [16], and promote neuronal stem cell differentiation by activating genes that inhibit self-renewal [15]. CncC—the *Drosophila* homolog of Nrf2—regulates metamorphosis through the activation of ecdysone biosynthetic and response genes in specific tissues [17]. During neuronal remodeling, CncC regulates dendrite pruning by activating gene expression of proteasomal subunits [18]. The molecular mechanisms and interacting partners that mediate the role of Nrf2 in the transcriptional regulation of development remain to be fully explored.

GATA family proteins are conserved transcription factors that are essential for the regulation of a broad range of early developmental programs [19]. Among the six major mammalian GATA factors, GATA4 regulates gene expression in cardiac progenitor cells during heart development [20]. Mutations in GATA4 are associated with congenital heart defects [21,22]. GATA4 regulates heart development through interaction with multiple cofactors, including NKX2-5 [22]. Molecular interactions between NKX2-5 and GATA4 have been identified in several studies [23,24]. It was found that the NKX2-5–GATA4 interaction facilitates the chromatin binding and transcriptional activities of GATA4 in COS-1 cells [25].

The *Drosophila pannier* (*pnr*) gene encodes the zinc finger transcription factor Pannier (Pnr), which is the homolog of human GATA4 [26,27,28]. Pnr is involved in multiple developmental processes during embryonic and imaginal development [29,30,31]. Mutation and shRNA depletion of Pnr alter *Drosophila* heart development [26,32]. Pnr binds to and activates cardiac developmental genes, including *Hand*, *Mef2*, and *mid*, in cooperation with the *Drosophila* NKX2-5 homolog Tinman (Tin) [33,34,35,36,37]. However, there are many genes that are regulated by Pnr beyond the genes coactivated with Tin, indicating that other factors cooperate with Pnr in transcriptional regulation.

Several lines of evidence suggest that GATA factors can regulate xenobiotic response genes. For example, GATA4 can directly bind to and activate the microsomal epoxide hydrolase gene *EPHX1* in HepG2 cells [38]. In *Drosophila*, GATA-binding sites were identified at the promoters of *Cyp6g1* [39]. Transcriptional assays of *Drosophila* embryos found that Pnr is required for the activation of some *P450* xenobiotic response genes, including *Cyp4p1*, *Cyp6a2*, and *Cyp6v1* [30]. Interestingly, upregulation of GATA4 was observed along with cardiac injury in rats treated with PM2.5 [40]. Nevertheless, the full-range molecular functions of GATA factors in the transcriptional regulation of xenobiotic/oxidative responses remain to be elucidated.

In this study, we identified a novel interaction between CncC and Pnr using the bimolecular fluorescence complementation (BiFC) assay—a unique imaging tool for probing protein complexes in living cells. We visualized CncC–Pnr BiFC complexes in the nuclei of *Drosophila* embryonic and salivary gland cells. Mapping the binding of the CncC–Pnr BiFC complex on the polytene chromosome revealed that the CncC–Pnr complex tends to bind to euchromatic interband regions, with a preference for loci that are not highly occupied by CncC or Pnr alone. Furthermore, genes within these loci were demonstrated to be coactivated by CncC and Pnr. These results suggest that CncC and Pnr interact with one another at specific genomic loci and coactivate transcription, revealing a new mechanism whereby CncC and Pnr co-regulate development and/or xenobiotic response.

## 2. Results and Discussion

### 2.1. CncC and Pnr Form Nuclear Complexes in Embryonic Mesodermal Cells

To determine whether CncC and Pnr interact in living cells, we applied the bimolecular fluorescence complementation (BiFC) assay [41] to visualize potential CncC–Pnr complexes in living tissues at different developmental stages of *Drosophila*. CncC and Pnr BiFC fusions (i.e., proteins fused to the N- and C-terminal fragments of YFP) were ubiquitously expressed using the UAS-GAL4 expression system with the *tub-GAL4* driver. BiFC fluorescence was detected in the nuclei of cells in embryos (Figure 1A), indicating that CncC and Pnr form complexes in the nucleus. We were unable to check the formation of CncC–Pnr BiFC complexes at later developmental stages in this experiment because the global overexpression of CncC, driven by the *tub-GAL4* driver, is lethal at the early first-instar larval (L1) stage [42].

### 2.2. CncC and Pnr Form Protein Complexes on Chromatin

To examine the detailed subcellular localization of the CncC–Pnr complex, we expressed CncC and Pnr BiFC fusions in polyploid salivary gland cells using the *Sgs3-GAL4* driver. In most cells, CncC–Pnr BiFC signals were detected exclusively on the polytene chromosome and enriched in euchromatic interband regions (Figure 1B). In a small portion of cells (<10%), accumulation of BiFC signals was also detected in the nucleoplasm, likely due to the overexpression of BiFC fusion proteins (Appendix A). YFP–CncC localized to both the nucleoplasm and polytene chromosome, whereas YFP–Pnr bound only to the polytene chromosome (Figure 1B and Figure 2A) [17]. The expression levels of BiFC fusions and YFP fusions were comparable at both the transcriptional and protein levels (Figure 1C and Figure 3A). Co-expression of YC–dKeap1 and YN–Pnr fusion proteins did not produce detectable fluorescence in salivary gland cells (Figure 1B), although both dKeap1 and Pnr bind to many interband regions on the polytene chromosome (Figure 2A) [17]. Therefore, the BiFC assay revealed specific molecular interactions between CncC and Pnr.

The *cnc* gene also codes for CncB—a chromatin-binding protein that is required for the normal embryonic development of *Drosophila* [43]. Compared with CncC, CncB shares the same DNA-binding domain but lacks the N-terminal dKeap1-interacting domains (Figure 1B). YFP–CncB fusion proteins bind to many interbands on the polytene chromosome [44], but co-expression of YC–CncB and YN–Pnr did not produce detectable fluorescence in salivary gland cells (Figure 1B). This suggests that Pnr interacts with the N-terminal domain of CncC. Selective BiFC complexes were formed by CncC and Pnr, but not by CncB or dKeap1 in combination with Pnr, indicating that CncC and Pnr form specific complexes on chromatin.

### 2.3. CncC and Pnr Form BiFC Complexes at Specific Genomic Loci

The polytene chromosome BiFC assay, for the first time, allowed a direct visualization of protein complexes at specific genomic loci [44,45]. To investigate the chromatin-binding specificity of the CncC–Pnr complex, we compared the genomic loci that were primarily occupied by CncC–Pnr BiFC complexes with those bound by CncC and Pnr separately on polytene chromosome spreads (Figure 2 and Appendix A). Genome-wide distributions of YFP–CncC and YFP–Pnr on polytene chromosomes were visualized by anti-GFP immunostaining. BiFC complexes could not be detected by immunostaining, since the anti-GFP antibody recognizes both YN and YC fragments; thus, the BiFC complex and separate YN and YC fusion proteins cannot be distinguished. Therefore, intrinsic CncC–Pnr BiFC fluorescence was visualized on polytene chromosome spreads that were prepared using an acid-free squash protocol, which preserved the live YFP fluorescence [46]. CncC–Pnr BiFC complexes and the separately expressed YFP–CncC and YFP–Pnr proteins all bind to a large proportion of interband regions (Figure 2A). Both BiFC complexes and YFP fusion proteins selectively occupy certain loci at significantly higher levels compared to their binding levels at adjacent chromatin regions. The loci that are strongly occupied by CncC–Pnr complexes differ from those occupied by YFP–CncC or YFP–Pnr that are expressed separately (Figure 2B). For example, the fluorescence intensity of CncC–Pnr BiFC was significantly higher at the 89E and 33B loci compared to adjacent loci (for example, 91A and 34A), while no predominant YFP–CncC or YFP–Pnr signals were detected at 89E and 33B (Figure 2B and Appendix A). On the other hand, loci that are highly occupied by CncC—such as the 74EF and 75B ecdysone-response puffs [17]—were not predominantly bound by CncC–Pnr BiFC or YFP–Pnr (Appendix A). Similarly, no predominant signals of CncC–Pnr BiFC or YFP–CncC were detected at some Pnr-favoring loci, such as 91A and 34A (Figure 2B). Taken together, these results suggest that YFP–CncC and YFP–Pnr bind to many chromatin loci independently of one another, and that CncC–Pnr BiFC complexes form at or bind to some loci that are not primarily occupied by CncC or Pnr. This result also verifies the polytene chromosome BiFC assay as a valuable tool to probe the binding of protein complexes at specific genomic loci.

Interestingly, both live imaging and immunostaining revealed strong YFP–Pnr signals at the polytene chromosome chromocenter—the centromeric region of the *Drosophila* genome (Figure 1B and Figure 2A). It was reported that a cell-cycle-dependent GATA factor, Ams2, binds to and controls the centromere in fission yeast through the regulation of histone H3 variant CENP-A [47], but the role of GATA factors at centromeres is largely unknown. It will be interesting to explore the potential function of Pnr/GATA in centromere architecture and pericentric heterochromatin in the future.

### 2.4. Regulation of Transcription by the CncC–Pnr Complex

To investigate whether CncC and Pnr regulated transcription in a way that was associated with their interactions at specific genomic loci, we examined gene transcription levels at the 89E and 33B loci in salivary glands that overexpressed CncC–Pnr BiFC, and we compared them with the levels of same transcripts regulated by overexpressed YFP–CncC, YFP–Pnr, or YFP (wild-type control) (Figure 3). Among the 21 genes at the 89E and 33B loci that were examined, three (*dKeap1*, *Or33b*, *Or33c*) were activated by CncC overexpression and two (*Pxd* and *Tom70*) were activated by Pnr overexpression (Figure 3B). None of these genes were significantly activated by both ectopic CncC and Pnr. Interestingly, seven other genes (*Glut3*, *Actn3*, *Dad*, *Ns1*, *Ada1-2*, *Ada1-1*, *Wdr81*) were specifically activated by the CncC–Pnr BiFC (by 3–15-fold) but not activated by the individually expressed CncC or Pnr (Figure 3B). dKeap1 transcription was activated by CncC (by around 9-fold) and was slightly activated by Pnr, whereas it was dramatically activated by the CncC–Pnr BiFC (by 30-fold) (Figure 3B). Therefore, the CncC–Pnr complex can specifically target and activate certain genes. The number of genes regulated by the CncC–Pnr complex remains unclear. The CncC–Pnr BiFC complex moderately occupied many chromatic loci. Thus, CncC and Pnr may interact with one another and co-regulate many genes.

To determine whether the CncC–Pnr interaction is required for the activation of all CncC or Pnr target genes, we examined the effects of CncC overexpression, Pnr overexpression, and CncC–Pnr BiFC for 19 genes that were identified as common targets of CncC and Pnr based on genome-wide transcriptional analyses of *cncC* or *pnr* mutations [48,49,50]. Some of these genes (i.e., *Ctr1A*, *Cyt-b5*, *Atx2*, *bbc*, *ifc*) were activated by CncC, and some (i.e., *Cpr*, *Ctr1A*, *fra*) were activated by Pnr, but only one (*Egfr*) was specifically activated by CncC–Pnr BiFC (Figure 3C). This result suggests that many genes, although regulated by both CncC and Pnr, are activated through mechanisms independent of the CncC–Pnr interaction. It has been previously demonstrated that CncC and Pnr can regulate transcription in cooperation with other cofactors, such as dKeap1 and Tin, respectively [35,44].

Among the genes that were targeted and activated by CncC–Pnr BiFC complexes at 89E and 33B, none of them were revealed as CncC- or Pnr-regulatory genes in previous genome-wide transcription analyses [30,48]. Given that both CncC and Pnr are critical for embryo development [17,32], we next investigated whether these genes were regulated by endogenous CncC and Pnr in embryos (Figure 3D). We compared the transcription levels of these genes in homozygous *cncC* or *pnr* null mutants and in relevant heterozygous embryos (*cnc^−/+^* or *pnr^−/+^*). Two independent null alleles for *cncC* (*cnc^K6^* and *cnc^K22^*) and *pnr* (*pnr^VX6^* and *pnr^1^*) were employed. The levels of *Glut3*, *Actn3*, and *Wdr81* transcripts reduced in both *cncC* and *pnr* mutants. *Ns1* and *Ada1-1* were significantly downregulated in at least one allele of *cncC* and *pnr* mutants. *Dad*, *dKeap1*, and *Ada1-2* were downregulated in *cncC* mutants (Figure 3D). Thus, most of the genes that were identified as CncC–Pnr BiFC targets in this study are likely real target genes of endogenous CncC and/or Pnr in embryos. Mapping BiFC complexes on polytene chromosomes has previously allowed us to reveal novel genes that are coactivated by dKeap1 and CncC [44]. This study provides another successful example supporting the polytene chromosome BiFC assay as a valuable tool for the identification of genes that are specifically targeted by a protein complex.

No classic Nrf2/CncC-binding element ARE was identified in genes targeted by the CncC–Pnr BiFC complex, indicating that CncC probably binds to these genes through the interaction with Pnr. Consistent with this hypothesis, the Cnc isoform lacking the N-terminal domain lost the interaction with Pnr (Figure 1B). Both genome-wide ChIP-Seq assays in murine cells and polytene chromosome staining in *Drosophila* revealed that Nrf2/CncC binds to many developmental genes in an ARE-independent manner [3,17]. The mechanisms that mediate the specific binding of Nrf2/CncC to these genes are not well understood. This study hints at a possible model where the binding of Nrf2 family proteins to some developmental genes is mediated by interaction with GATA factors (Figure 4B). In support of this model, GATA factors have been previously demonstrated as pioneer factors that can open local chromatin and facilitate the recruitment of other transcription factors [19]. The complete molecular mechanism whereby CncC/Nrf2 and Pnr/GATA4 factors interact and co-regulate transcription remains to be explored in *Drosophila* and mammalian model systems.

### 2.5. Genetic Interaction between CncC and Pnr

Most of the CncC–Pnr co-target genes identified in this study are developmental genes, indicating that CncC and Pnr may co-regulate development. Therefore, we tested the genetic interaction between CncC and Pnr during *Drosophila* development. CncC overexpression driven by *tub-GAL4* caused severe embryonic lethality, with less than 10% of animals surviving to the early first-instar larval (L1) stage (Figure 4A). Depletion of Pnr by RNAi reduced the viability at third-instar larval (L3), pupal, and adult stages. Combinatory expression of *pnr-RNAi* significantly rescued the survival ratio of CncC-overexpressing animals to ~57% at the L1 stage, ~27% at the L3 stage, and ~16% at the pupal stage (Figure 4A). Depletion of Pnr by RNAi had no significant effects on the transcript or protein levels of CncC (Figure 3D and Appendix A). These results suggests that CncC and Pnr function in the same pathway in the regulation of development, likely mediated by the developmental genes that are controlled by the CncC–Pnr complex.

It remains to be explored how the CncC–Pnr interaction regulates developmental genes and programs. As an oxidative and xenobiotic response factor, Nrf2 is globally expressed in all cell types [51]. Previous studies also showed that the *cncC* transcript isoform was expressed in all cells during embryogenesis, and CncC proteins were detected in many tissues [17,44,52,53]. Although the basal expression levels of CncC/Nrf2 are very low in unstressed conditions, CncC/Nrf2 can directly target and regulate developmental genes [13,14,15,16,17]. Pnr regulates developmental genes and processes in multiple tissues, such as embryonic mesoderm and imaginal discs [29,30,31]. Combined with their genetic interaction detected in this study, we hypothesize that CncC and Pnr form complexes and regulate developmental genes in some cell types. Our current study provides evidence in support of further functional studies of the CncC–Pnr complex in specific tissues during *Drosophila* development.

Taken together, our study revealed novel molecular and genetic interactions between the CncC xenobiotic response factor and the Pnr transcription factor in *Drosophila*. CncC and Pnr interact at specific genomic loci where they mediate the activation of target genes, suggesting a model whereby the CncC–Pnr complex regulates development through binding and activating specific developmental genes (Figure 4B). It was revealed that Nrf2/CncC can regulate developmental genes in a redox-independent manner, but the underlying molecular mechanisms remain to be explored. This study identified GATA4 as a novel cofactor of Nrf2 in the regulation of development.

Given the role of both Nrf2 and GATA4 in cardiac development, our study also hints at potential crosstalk between classic oxidative/xenobiotic response signaling and the transcription-regulatory network controlling cardiac development. It is well established that Nrf2 can protect the cardiovascular system through the regulation of redox hemostasis in blood vessels, blood cells, and the myocardium [54]. The pathogenic roles of Nrf2 in cardiovascular diseases, including atherosclerosis and heart failure, have also been revealed by several studies [10,54]. Despite the roles of Nrf2 in cardiovascular specific cell protection, the function of Nrf2 in cardiovascular development remains unclear. The discovery of the Nrf2–GATA4 interaction suggests the presence of a novel molecular regulatory network that underlies cardiac development, providing more insight into the roles of Nrf2 and GATA factors in development and diseases.

## 3. Methods

### 3.1. Drosophila Stocks

Plasmids encoding CncC and Pnr fused to YFP were constructed using the pUAST vector and microinjected in the *w^1118^* background. *UAS-YC-CncC*, *UAS-YC-CncB*, and *UAS-YC-dKeap1* strains were obtained from Osamu Shimmi. *UAS-YN-Pnr* was provided by Rolf Bodmer. *cnc^K6^* and *cnc^K22^* were obtained from Dirk Bohamn. *Sgs3-GAL4*, *tub-GAL4*, *pnr^VX6^*, *pnr^1^*, *pnr-RNAi*, and *UAS-YFP* were obtained from the Bloomington Stock Center. All stocks were maintained at 25 °C according to the standard protocol. Null mutations of *cncC* (*cnc^K6^*, *cnc^K22^*) and *pnr* (*pnr^VX6^*, *pnr^1^*) are embryonically lethal and, thus, were maintained in combination with the *TM6, Tb, Sb, Hu, e, Dfd-YFP* balancer. To isolate null mutants, embryos were collected on apple-juice plates and aged for 20–22 h at 25 °C, and then sorted based on the *Dfd-YFP* marker under a Leica MZ10 F fluorescence stereomicroscope. In the genetic interaction assay, *tub-GAL4/TM6* females were crossed with *UAS-YFP-CncC*, *UAS-pnr-RNAi*, or *UAS-YFP-CncC;UAS-pnr-RNAi* males. F1 flies without *tub-GAL4* were used as controls and identified based on *Dfd-YFP* (in the first instar stage) and *Tb* (in pupal and adult stages) markers on the *TM6* balancer.

### 3.2. Live Imaging of BiFC Fluorescence in Salivary Gland Cells

To express YFP or BiFC fusion proteins in salivary glands, UAS transgenic lines or double-transgenic lines were crossed with *Sgs3-GAL4*. Salivary glands were isolated from F1 early wandering third-instar larvae and stained with 10 µg/mL Hoechst 33258 in PBS. After brief washing, they were mounted in PBS and immediately imaged using a Nikon Eclipse 80 fluorescence microscope with a SPOT Insight 4 MP color digital camera. The signals were pseudo-colored and merged in the RGB color space.

### 3.3. Imaging of BiFC Fluorescence on Polytene Chromosomes

Double-transgenic lines expressing BiFC fusion proteins were crossed with *Sgs3-GAL4*. F1 larvae were maintained at 21 °C to enhance the polyploidy of salivary gland cells. Salivary glands were isolated from early wandering third-instar larvae. Polytene chromosome spreads were prepared using an acid-free squash technique to avoid quenching of the YFP and BiFC fluorescence [46]. One pair of dissected salivary glands was incubated in freshly prepared 2% paraformaldehyde in Brower’s Fixation Buffer (0.15 M PIPES, 3 mM MgSO_4_, 1.5 mM EGTA, 1.5% NP40, pH 6.9) for 3 min, in PBST (PBS + 0.2% Triton X-100) for 3 min, and in 50% glycerol for 5 min. The salivary glands were then squashed in 10 μL of 50% glycerol and stained with 0.2 µg/mL Hoechst 33258 in PBS before mounting in 80% glycerol with 10 mM Tris, pH 9.0. Images were acquired on a Nikon AX-R confocal microscope. BiFC signals were visualized using 504 nm excitation and 542 nm emission wavelengths.

### 3.4. Immunostaining

Polytene chromosome spreads were prepared from the salivary glands of early-wandering third-instar larvae using conventional squash and immunostaining protocols [46]. The antibodies used for immunostaining were anti-GFP (1:200, Fitzgerald Industries, Acton, MA) and Alexa Fluor 488 conjugated goat anti-rabbit secondary antibody (1:2000, Invitrogen, Waltham, MA, USA). The specificities of the antibodies had been tested and verified (Appendix A). The stained samples were mounted in VectaShield (Vector Laboratories, Newark, CA, USA) and imaged using a Nikon Eclipse 80 fluorescence microscope with a SPOT Insight 4 MP color digital camera.

### 3.5. Western Blotting

Ten pairs of salivary glands were homogenized in 50 μL of ice-cold IP buffer (20 mM Tris-HCl pH 8.0, 0.2% Triton X-100, 150 mM NaCl, 5 mM EDTA, 2 mM NaVO3, 1 mM PMSF, and 1.5 μg/mL aprotinin). Proteins were separated using a 10% Tris-glycine gel and transferred to nitrocellulose membranes (Bio-Rad, Hercules, CA, USA). Membranes were blocked with 5% milk in TBST (TBS + 0.1% Tween-20) and then probed with primary antibodies against GFP (1:1000, Fitzgerald Industries, Acton, MA, USA), CncC [17], and tubulin (12G10, 1:500, Developmental Studies Hybridoma Bank, Iowa City, IA, USA), followed by 1:3000 HRP-conjugated goat anti-rabbit or goat anti-mouse secondary antibodies (Bio-Rad, Hercules, CA, USA). The membranes were developed using ECL reagents (GE Healthcare, Chicago, IL, USA) and then exposed to X-ray film (AGFA, Mortsel, Belgium).

### 3.6. Transcript Quantitation by RT-qPCR

mRNAs from 10 pairs of salivary glands dissected from early-wandering third-instar larvae in PBS, prepared using DEPC water, were extracted using the RNeasy kit (Qiagen, Germantown, MD, USA). Isolated mRNA was treated with RQ1 RNase-Free DNase (Promega, Madison, WI, USA) and reverse-transcribed using the iScript^TM^ cDNA Synthesis kit (Bio-Rad, Hercules, CA, USA). Real-time qPCR was performed using SYBR Green I Master (Genesee Scientific, San Diego, CA, USA) in an Eppendorf realplex Mastercycler. In each qPCR experiment, two technical repeats were applied for each sample. The relative transcript levels were calculated by assuming that they were proportional to 2^−Cp^ and normalized to the levels of *Rp49* transcripts. Primer sequences were designed using Universal ProbeLibrary (Roche, Indianapolis, IN, USA) or PrimerQuest (IDT, Coralville, IA, USA) and are listed in Appendix A. Statistical analyses for the significance of differences in relative transcript levels were evaluated using two-way ANOVA based on three separate experiments.

## Figures and Tables

**Figure 1 cells-12-00938-f001:**
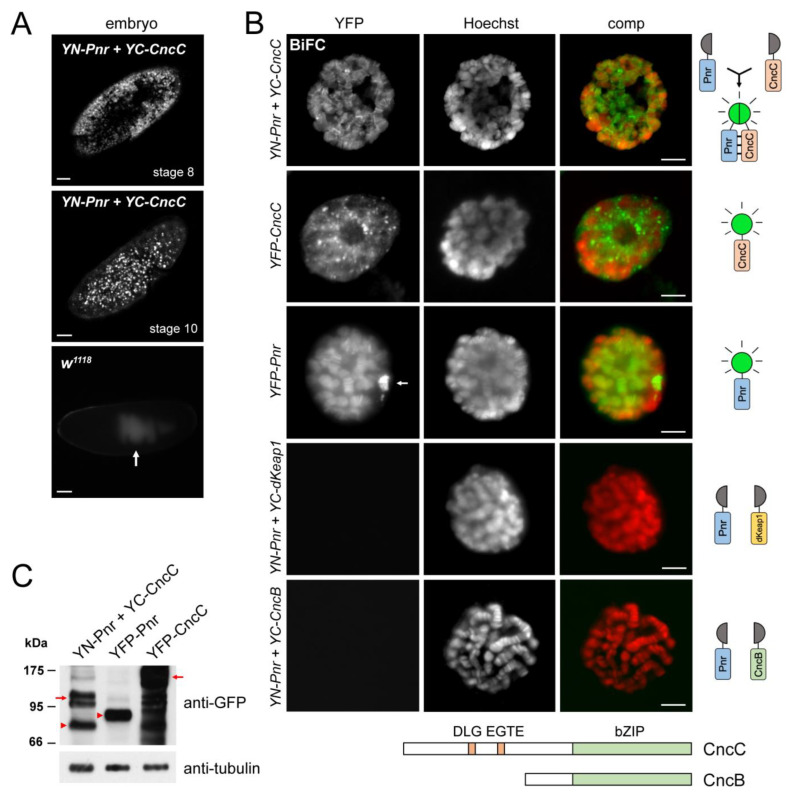
Visualization of CncC–Pnr BiFC complexes in live cells: (**A**) Formation of CncC–Pnr BiFC complexes in the embryo. YC–CncC and YN–Pnr fusion proteins were expressed using the *tub-GAL4* driver. Intrinsic BiFC fluorescence was imaged in dechorionated live embryos at the stages indicated. The fluorescence represented specific BiFC complexes as indicated by comparison to wild-type (*w^1118^*) embryos with the same exposure. Arrow: autofluorescence produced by yolk. Scale bars: 50 μm. (**B**) Visualization of CncC–Pnr BiFC, YFP–CncC, and YFP–Pnr in salivary gland cells. Fusion proteins described on the left were expressed using the *Sgs3-GAL4* driver. Salivary glands were dissected and stained with Hoechst (red). BiFC fluorescence or intrinsic YFP fusion proteins (green) were imaged. Diagrams on the right depict the CncC–Pnr BiFC complex in comparison with YFP fusion proteins. As controls, YC–dKeap1 or YC–CncB were co-expressed with YN–Pnr, and no BiFC signal was detected. The bottom diagram depicts the comparison of the CncC and CncB protein domains. Arrow: strong YFP–Pnr signal at the chromocenter. Scale bars: 10 μm. (**C**) Expression levels of fusion proteins. Extracts of salivary glands that expressed CncC–Pnr BiFC (YN–Pnr + YC–CncC), YFP–CncC, or YFP–Pnr were analyzed by Western blotting using antibodies against GFP and tubulin (loading control). Marker sizes (kDa) are indicated on the left. CncC and Pnr fusion proteins are indicated by arrows and arrowheads, respectively. See the lower exposure in Appendix A.

**Figure 2 cells-12-00938-f002:**
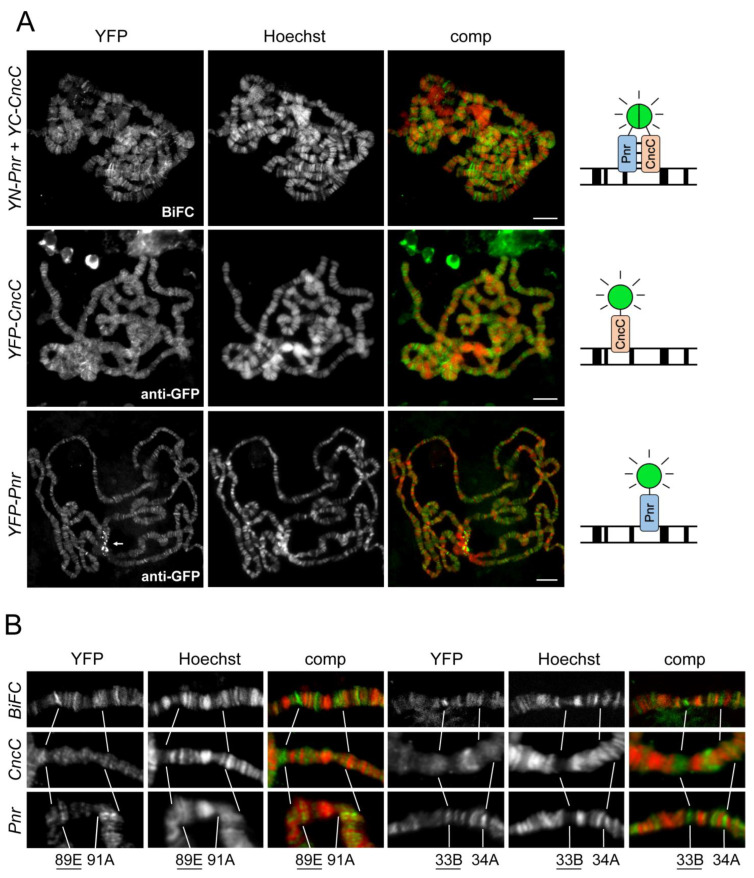
Visualization of CncC–Pnr BiFC complexes at chromatin loci: (**A**) Visualization of CncC–Pnr BiFC, YFP–CncC, and YFP–Pnr binding on polytene chromosomes. The fusion proteins indicated on the left were expressed in salivary glands using the Sgs3-GAL4 driver. CncC–Pnr BiFC fluorescence (green) was visualized intrinsically, while YFP–CncC and YFP–Pnr bindings were visualized by anti-GFP immunostaining (green). Polytene chromosome spreads were counterstained with Hoechst (red). The diagrams on the right depict the polytene chromosome BiFC assay of the CncC–Pnr complex in comparison with analyses of YFP fusion proteins. Arrow: strong YFP–Pnr signal at the chromocenter. Scale bars: 10 μm. (**B**) Comparison of CncC–Pnr BiFC, YFP–CncC, and YFP–Pnr bindings at selected loci. Fragments of polytene chromosome spreads prepared as described in (**A**) were aligned and compared at the loci indicated below. The loci that were highly occupied by CncC–Pnr BiFC complexes are underlined. Adjacent control loci 91A and 34A were highly occupied by YFP–Pnr. For the relative entire chromosome arm 3R containing the 89E fragments shown here, refer to Appendix A. See Appendix A for additional examples. See Appendix A for the comparison of CncC–Pnr BiFC, YFP–CncC, and YFP–Pnr bindings at CncC-favoring loci.

**Figure 3 cells-12-00938-f003:**
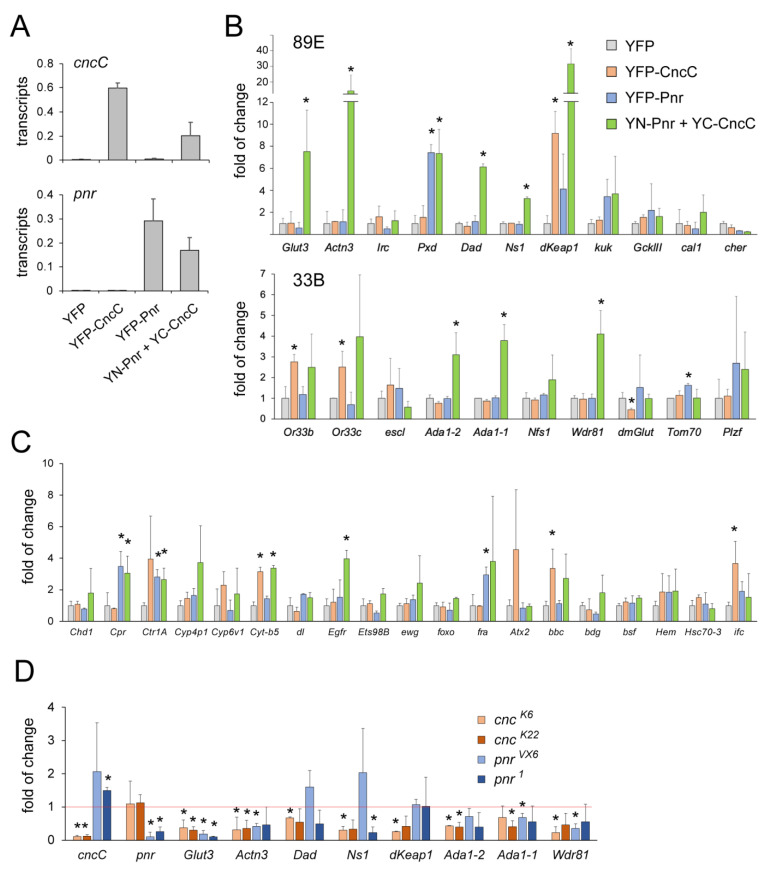
Co-regulation of transcription by CncC and Pnr: (**A**) *Expression levels of CncC and Pnr fusions.* The levels of *cncC* and *pnr* transcripts in salivary glands that expressed YFP (control), YFP–CncC, YFP–Pnr, or CncC–Pnr BiFC fusions using *Sgs3-GAL4* were measured by RT-qPCR. Transcript levels were normalized by the levels of *Rp49* transcripts. The data show the mean and standard deviation of results from three separate experiments. (**B**) *Effects of ectopic CncC, Pnr, and CncC–Pnr BiFC on transcription of genes at loci that are highly bound by CncC–Pnr BiFC complexes.* Transcript levels of genes (listed below) at the 89E and 33B loci in salivary glands that that expressed YFP (control), YFP–CncC, YFP–Pnr, or CncC–Pnr BiFC fusions using *Sgs3-GAL4* were measured by RT-qPCR. All transcript levels were normalized by the levels of *Rp49* transcripts. To facilitate comparison, the transcript levels in cells expressing ectopic fusion proteins were normalized to the levels of transcripts in control cells (*Sgs3* > *YFP*). Standard deviations were calculated based on three independent experiments (*, *p* < 0.05). (**C**) *Effects of ectopic CncC, Pnr, and CncC–Pnr BiFC on other genes.* Transcription levels of the genes labeled below were measured in the same salivary gland samples and analyzed in the same way as described in (**B**). Standard deviations were calculated based on three independent experiments (*, *p* < 0.05). (**D**) *Effects of CncC and Pnr knockouts on transcripts at loci predominantly bound by CncC–Pnr BiFC complexes.* Transcription levels of the genes labeled below, in embryos of *cncC* or *pnr* null mutations (*cnc^K6/K6^*, *cnc^K22/K22^*, *pnr^VX6/VX6^*, or *pnr^1/1^*) and relevant heterozygous control embryos (*cnc^K6/+^*, *cnc^K22/+^*, *pnr^VX6/+^*, or *pnr^1/+^*), were measured by RT-qPCR. All transcript levels were normalized to the levels of *Rp49* transcripts. To facilitate comparison, the transcript levels in null embryos were normalized to the levels of transcripts in control embryos. Standard deviations were calculated based on three independent experiments (*, *p* < 0.05).

**Figure 4 cells-12-00938-f004:**
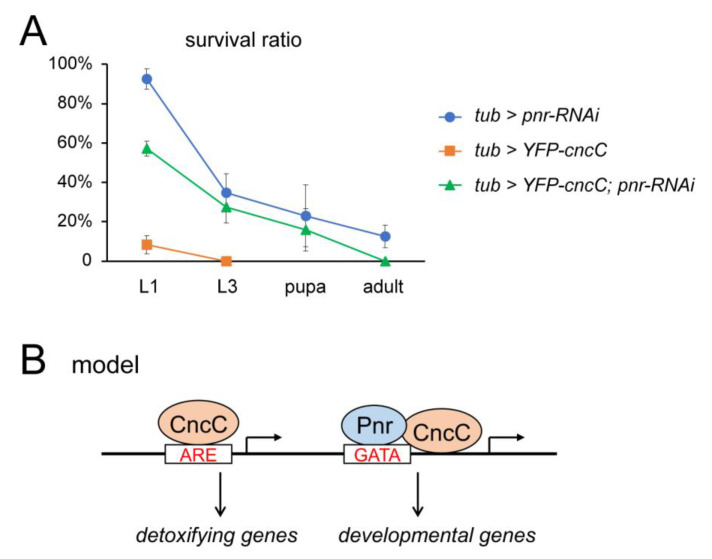
CncC and Pnr function in the same genetic pathway: (**A**) Genetic interaction between cncC and pnr mutants. Numbers of Pnr-knockdown flies (*tub-GAL4/pnr-RNAi*), CncC-overexpressing flies (*UAS-YFP-cncC/+; tub-GAL4/+*), and the double mutant combining CncC overexpression and Pnr knockdown (*UAS-YFP-cncC/+; tub-GAL4/pnr-RNAi*) were counted at the stages listed below. Survival ratios were calculated by normalization to control flies without *tub-GAL4* drivers. The control files were cultured in parallel and identified based on *Dfd-YFP* and *Tb* markers on the *TM6* balancer. Error bars represent the standard deviation, based on two independent experiments with more than 50 flies counted. (**B**) Model of CncC–Pnr interaction. As a xenobiotic and oxidative response factor, Nrf2/CncC activates detoxifying genes through binding to the ARE element. The CncC–Pnr complex binds to specific chromatin loci and activates developmental genes, indicating a potential novel developmental function of Nrf2 in cooperation with a GATA factor.

## Data Availability

Not applicable.

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
