# Peer review of "Determination of Complex Formation between Drosophila Nrf2 and GATA4 Factors at Selective Chromatin Loci Demonstrates Transcription Coactivation"

_cells, 2023, doi:10.3390/cells12060938_

Round 1

Reviewer 2 Report

In this manuscript, the authors ectopically express two different transcription factors (CncC – Drosophila Nrf2 and Pnr – Drosophila GATA4) tagged with different complementary portions of YFP to demonstrate through BiFC fluorescence that the transcription factors colocalize in embryos and on polytene chromosomes of the third instar salivary gland.  They then quantitate expression of genes that map to the regions on the salivary gland polytene chromosomes where they observe YFP BiFC using RT-qPCR and show that many genes change in expression when one or both of the two TFs are overexpressed or are missing.  As final support for the idea that the two proteins may functionally interact, they show that knockdown of pnr by RNAi can rescue the viability loss associated with overexpression of cncC.  The authors conclude that the two proteins work together to activate gene expression during cellular stress and during development.

This manuscript is clearly written and the images are (mostly) high quality.  However, the data do not currently support the major conclusions of the manuscript, as detailed below.  Indeed, more experiments would need to be done to support the major conclusions of the manuscript.  For example, is there any evidence the cncC and pnr are normally expressed in the same cells either in embryos or in larval tissues.  If not, all of this is moot.  If so, one needs to then know if the two proteins are co-expressed (normally – not with transgenes) and if the two proteins are ever found in nuclei together.  If not, again, the findings from this work do not have a lot of meaning beyond (perhaps) showing the the BiFC works.

Concerns:

Figure 1A.  This staining looks like the yolk autofluorescence that is visible in all embryos in certain channels.  The authors need to provide some more compelling evidence (including a multitude of controls) that the staining is what they think it is.  Controls should include the fluorescence observed with

(1)   Each TF alone tagged with GFP

(2)   Each TF tagged with GFP expressed in the same cells as a non-tagged version of the other TF

(3)   No expression of either transgene used in the BiFC

Figure 1B.  The same controls need to be included for the SG staining, although each protein alone is shown already.

Supplementary Figure 1A also needs many of the same controls.  The staining of the embryo in the right panel of A is exactly where one would expect to see yolk autofluorescence.

Supplementary Figure 1C.  The BiFC signals should span across the chromosome if they are real – they would not be the little blobs of YFP on one side of the chromosomes (I think) the authors are showing. 

Figure 2A.  Some additional controls are needed here as well.  What does secondary alone look like when used on SG polytene chromosomes?  What does the staining look like on WT chromosomes that are not expressing the GFP-tagged transcription factors?

Figue 2B. It would be good to have some independent confirmation that the regions of polytene chromosomes shown are indeed what is claimed by the authors.  Perhaps a less enlarged version showing the ends of the relevant chromosomes with the regions shown in the figure boxed out so that the readers can have more confidence in the assignments by the authors.  Also, assuming that the chromosomal assignments are correct, one would conclude that each of these proteins binds to some sites only in the presence of the other – in which case they need to show this by expressing the GFP-tagged version of one TF in the same cells as an untagged version of the other TF.  Staining of these genotypes should include the bands that are positive by Bi-FC.

Figure 3.  It's not clear how many biological and technical replicates were used in the RT-qPCR experiments.

Figure 3D. Looking at gene expression in embryos homozygous for a mutation that has been maintained over a balancer chromosome for many years is not ideal.  Other mutations may have accumulated on these chromosomes and these other mutations may affect transcript levels.  The authors need to cross independent null alleles of these genes to each other to do these experiments.  Or they need to cross the null allele to a small deficiency removing the transcription factor.

Reviewer 3 Report

In this manuscript, by using bimolecular fluorescence complementation (BiFC) assay, Neidviecky and Deng aimed to detect the interaction between CncC and Pnr and found the binding loci of CncC and Pnr complexes in the salivary gland of Drosophila. Furthermore, the authors found that the CncC and Pnr together regulate the transcription of some genes within the binding loci in a CncC-Pnr complex-dependent manner. Overall, the authors’ findings are interesting, and the results are convincing. However, I believe some minor changes are necessary to better support the authors' conclusions.

Comments:

1. In figure 1C, why is the size of YFP-CncC much larger than YC-CncC? Can the authors provide a better result for western blot? Because the YFP-CncC lane cannot be seen clearly.

2. Line 164 is confusing, need to rephrase to emphasize that the GFP antibody has to recognize when both YC- and YN- exist.

3. In figure 3D, why didn’t the authors test the genes in the salivary gland? Can’t these mutant flies survive in the larvae stage? Because the authors did test genes’ transcription in the salivary gland upon overexpression of YFP-CncC, YFP-Pnr, and YC-CncC YN-Pnr.

4. Typo “ccnC” is in many figures.

5. In figure 4A, can the authors test the protein level of CncC after Pnr Knockdown? Even though the authors showed RT-qPCR results of CncC upon Pnr Knockdown, it is possible that Knockdown Pnr causes degradation of CncC.

Round 2

Reviewer 1 Report

In the revised version of their manuscript, the author have addressed some of my previous criticisms, in particular by showing an additional example of chromosomal stainings for each locus. (Note: The revised version only included the new supplementary figures but did not include the figures of the main body). I am satisfied with some of the changes but there are some issues that still need to be improved upon. Because there are no page or line numbers, below I will cite the relevant text passages.

1.     Abstract: “GATA4 is a conserved GATA family transcription factor that is essential for cardiac development”. Please modify to: “essential for cardiac and dorsal epidermal development”.

2.     Omit current Fig. 1A and replace with current Fig. S1A, B, which is more convincing (see also previous critique of reviewer 1).

3.     “were unable to visualize CncC-Pnr BiFC complexes at later developmental stages”. Define which later stages (Larval? Late embryos, e.g., st. 16, should be doable).

4.     Fig. S2A, 33B locus: comp panel is duplicated, Hoechst panel is missing.

5.     “Among genes that are targeted and activated by Tin-Pnr BiFC complexes at 89E and 33B,…”. It is still unclear where this claim of Tin-Pnr BiFC data comes from (not from the references cited) and what Tin has to do with current analysis.

6.     “Ns1 and Ada1-1 were significantly downregulated in at least one allele of cncC and pnr mutants.” The previous figure only showed the data from one allele of each gene. According to the figure legend, these data should now be included in Fig. 3D. However, as stated above, the figures were not included in the revised manuscript, so I cannot judge the results (This issue also touches a criticism of reviewer 1).

7.     “Given the role of both CncC and GATA4 in the cardiac development, our study also illuminates a potential crosstalk…”. Since this aspect is speculative, “illuminates” is the wrong word here. Please replace with “hint at”. Omit “the” before “cardiac”.

8.     Methods: “Two technical repeats were applied for each sample.” “were evaluated using two-way ANOVA based on two to three separate experiments”. Inconsistent and vague information. Three independent experiments should be the minimum, especially in light of the large error bars in Fig. 3.

9.     Correct various language errors in newly modified text.

Author Response

Again, we are grateful for the insightful feedback on our manuscript. We have carefully addressed the reviewer comments in the revised manuscript and provided responses to each of the criticisms given in a point-by-point manner. The major changes in the revised manuscript are shown in red front.

Please see below for the “point-by-point” responses to each of the criticisms (Italic) raised by the reviewer.

  1. Abstract: “GATA4 is a conserved GATA family transcription factor that is essential for cardiac development”. Please modify to: “essential for cardiac and dorsal epidermal development”.

Thanks for the suggestion. We have modified it.

  1. Omit current Fig. 1A and replace with current Fig. S1A, B, which is more convincing (see also previous critique of reviewer 1).

According to this suggestion, we have moved the embryo BiFC control images from Figure S1A into Figure 1A.

  1. “were unable to visualize CncC-Pnr BiFC complexes at later developmental stages”. Define which later stages (Larval? Late embryos, e.g., st. 16, should be doable).

We apologize for this confusion and appreciate the suggestion. Visualization of the CncC-Pnr BiFC complex in embryos was to confirm complex formation and protein interaction in the living organism, the embryo stage at which the complex is visualized is of little importance for this purpose. Since that the BiFC fusions are expressed using tub-GAL4, this study was not able to examine endogenous CncC-Pnr complexes in embryos. Therefore, no particular stage of embryo was chosen over another. Additionally, the reason we can’t check CncC-Pnr BiFC in later developmental stages is because animals overexpressing CncC driven by UAS-GAL4 are lethal at the early L1 stage. We have modified the wording to clarify this in the text.

  1. Fig. S2A, 33B locus: comp panel is duplicated, Hoechst panel is missing.

Thanks for catching this error, we have replaced the duplicated image with the correct Hoechst image.

  1. “Among genes that are targeted and activated by Tin-Pnr BiFC complexes at 89E and 33B,…”. It is still unclear where this claim of Tin-Pnr BiFC data comes from (not from the references cited) and what Tin has to do with current analysis.

We deeply apologize for this mistake. We have corrected it to “CncC-Pnr BiFC”.

  1. “Ns1 and Ada1-1 were significantly downregulated in at least one allele of cncC and pnr mutants.” The previous figure only showed the data from one allele of each gene. According to the figure legend, these data should now be included in Fig. 3D. However, as stated above, the figures were not included in the revised manuscript, so I cannot judge the results (This issue also touches a criticism of reviewer 1).

Thanks for pointing this out. There may have been an error in figure uploading. Data from the additional alleles cnc[k22] and pnr[1] were shown as dark orange and dark blue columns in the updated Figure 3D.

  1. “Given the role of both CncC and GATA4 in the cardiac development, our study also illuminates a potential crosstalk…”. Since this aspect is speculative, “illuminates” is the wrong word here. Please replace with “hint at”. Omit “the” before “cardiac”.

Thank you for helping us revise the wording here, we have replaced "illuminates" with "hints to".

  1. Methods: “Two technical repeats were applied for each sample.” “were evaluated using two-way ANOVA based on two to three separate experiments”. Inconsistent and vague information. Three independent experiments should be the minimum, especially in light of the large error bars in Fig. 3.

Thanks for pointing out this confusion. The “Two technical repeats” is routinely used for qPCR experiments, which was listed here upon the request from one reviewer. We have clarified this as “In each qPCR experiment, two technical repeats were applied for each sample”. Three biological repeat experiments were conducted for all the genes in Figure 3 (we have added one more repeat to genes in Figure 3D and modified the methods to reflect this). The multiple repeats can reduce the p-value in statistical analyses but did not reduce the error bars representing standard deviation because the variance remained similar with the additional repeats.

  1. Correct various language errors in newly modified text.

We have carefully proofread the manuscript and corrected errors.

Reviewer 2 Report

A subset of my concerns have been partially addressed including

1.  the authors reassuring me that they have the skill set to identify regions of salivary gland polytene chromosomes even if they cannot trace the chromosomal regions to the ends of the chromosomes, which is where most people who can do this at least start for mapping purposes.  They must have had extensive training in this arena.

2. that the authors have looked at more than one loss of function allele of their two major players for changes in candidate target gene expression.  Nonetheless, it is not enough for them to simply tell me they have done this - the data from these other alleles needs to be in their figures.  Since I have gotten no new figures associated with this manuscript (other than some additional images in supplements)- I'm assuming they did not add that data to any figures.

3.  The authors have responded to my seeing "blobs of staining" not actually on the polytene chromosomes, but adjacent to them.

Remaining concerns:

1.  The authors conclude that the presence of TF1 affects the binding specificity of TF2 - they say they cannot do the appropriate controls because it would take time.  Yes, doing the appropriate controls takes time.  It is not a reason for not doing them.

2.  The authors address the issue of whether these two transcription factors are ever expressed in the same cells by saying:  "As xenobiotic response factors, CncC/Nrf2 proteins are globally expressed in all types of cells. Pnr regulates developmental genes and processes in multiple tissues such as the embryonic mesoderm and imaginal discs. Therefore, we hypothesize that endogenous CncC and Pnr co-exist in many cells and coregulate developmental genes through forming complexes. We have added these explanations into the discussion and provided references for the overlapping expression patterns of CncC and Pnr." The problem is that if you look at the expression data that are available for these genes in the BDGP database, there is no embryonic cell type where both genes are expressed.  Nor is the cnc gene expressed in larval salivary glands.  So, it is challenging to assess how meaningful the interactions observed with ectopic expression are with regards to what these proteins are normally do.

Author Response

Again, we are grateful for the insightful feedback on our manuscript. We have carefully addressed the reviewer comments in the revised manuscript and provided responses to each of the criticisms given in a point-by-point manner. The major changes in the revised manuscript are shown in red front.

Please see below for the “point-by-point” responses to each of the criticisms (Italic) raised by the reviewers.

A subset of my concerns have been partially addressed including

  1. the authors reassuring me that they have the skill set to identify regions of salivary gland polytene chromosomes even if they cannot trace the chromosomal regions to the ends of the chromosomes, which is where most people who can do this at least start for mapping purposes.  They must have had extensive training in this arena.

Thank you for the understanding. According to your suggestion, larger images of the whole chromosome 3R containing the 89E region shown in Figure 2B (left) were shown in Figure S1A (Figure S2 was reorganized and the original Figure S2C was moved to Figure S3). The chromosome with CncC-Pnr BiFC signals loses the end but can still be identified based on the binding pattern.

  1. that the authors have looked at more than one loss of function allele of their two major players for changes in candidate target gene expression.  Nonetheless, it is not enough for them to simply tell me they have done this - the data from these other alleles needs to be in their figures.  Since I have gotten no new figures associated with this manuscript (other than some additional images in supplements)- I'm assuming they did not add that data to any figures.

Data from the additional alleles cnc[k22] and pnr[1] were shown as dark orange and dark blue columns in the updated Figure 3D. We are sorry that there must have been an error uploading the revised figure. Please check the updated figures.

  1. The authors have responded to my seeing "blobs of staining" not actually on the polytene chromosomes, but adjacent to them.

Remaining concerns:

  1. The authors conclude that the presence of TF1 affects the binding specificity of TF2 - they say they cannot do the appropriate controls because it would take time.  Yes, doing the appropriate controls takes time.  It is not a reason for not doing them.

We are grateful for your insightful thoughts on our manuscript. However, we have decided not to conduct these experiments at this time. Our current data is solid and sufficient to support our main conclusion, which is that Nrf2 and GATA4 form complexes at specific genomic loci. While the proposed experiments may be valuable for addressing deep molecular mechanisms in the future, they are not critical for supporting our current conclusion.

We do agree that we can’t claim that “CncC and Pnr affect the binding specificity of each other”, without additional data showing that one protein affects the chromatin binding of the other. Therefore, we carefully tuned down the words in the text. For examples: “This study suggests a model that GATA factors can direct the binding of Nrf2 family proteins to developmental genes” was revised to “This study hints a possible model where the binding of Nrf2 family proteins to some developmental genes is mediated by interaction with GATA factors”; The speculations “It is possible that the interaction between CncC and Pnr can enhance the binding and transcriptional activity of each other” and “The CncC-Pnr interaction likely directs the chromatin binding specificity and transcriptional activity of CncC and Pnr” were deleted; We also mention that “The complete molecular mechanism whereby CncC/Nrf2 and Pnr/GATA4 factors interact and co-regulate transcription remains to be explored”. We appreciate your understanding of our decision and hope that our explanation adequately addresses your concerns.

  1. The authors address the issue of whether these two transcription factors are ever expressed in the same cells by saying:  "As xenobiotic response factors, CncC/Nrf2 proteins are globally expressed in all types of cells. Pnr regulates developmental genes and processes in multiple tissues such as the embryonic mesoderm and imaginal discs. Therefore, we hypothesize that endogenous CncC and Pnr co-exist in many cells and coregulate developmental genes through forming complexes. We have added these explanations into the discussion and provided references for the overlapping expression patterns of CncC and Pnr." The problem is that if you look at the expression data that are available for these genes in the BDGP database, there is no embryonic cell type where both genes are expressed.  Nor is the cnc gene expressed in larval salivary glands.  So, it is challenging to assess how meaningful the interactions observed with ectopic expression are with regards to what these proteins are normally do.

Thank you for thinking about the implications of our study critically. It has been found that the Nrf2 oxidative/xenobiotic response factor is expressed in all cell types, although its basal expressing level is very low due to the Keap1-mediated proteasomal degradation (de la Vega et al., NRF2 and the Hallmarks of Cancer. Cancer Cell 2018). The Drosophila cnc gene produces three protein isoforms, CncA, CncB, and CncC. Among them, only CncC is a homolog to Nrf2 because it contains the Keap1-interaction domains. CncB is a transcription factor that regulates embryonic development (Veraksa et al., Development 2000). Previous studies showed that the cncC transcript isoform is expressed in all cells during embryogenesis at levels that vary at different stages but much lower than those of cncB (McGinnis et al., Development 1998). Therefore, the embryonic expression pattern of cnc in BDGP database is CncB, and the low levels of CncC are ignored. Many studies have shown that the basal level of Nrf2/CncC, although very low, still regulates developmental transcription. For example, we found that CncC does express in larval salivary glands and regulates ecdysone response genes (Deng & Kerppola, 2013), regardless of the apparent no cnc expression reported in the BDGP database. To explain this, we added the above statements and references into the text.

We greatly appreciate your thoughtful feedback and suggestions regarding our manuscript. We understand that our study may raise further questions that need to be addressed in future research. Here, we have made every effort to ensure the accuracy and validity of our data. Additionally, we believe that our results make a significant contribution to the field and meet the standards of a Short Communication in Cells.